# Treatment in Latent Tuberculosis Uveitis—Is Immunosuppression Effective or Is Conventional 3- or 4-Drug Antituberculosis Therapy Mandatory?

**DOI:** 10.3390/jcm11092419

**Published:** 2022-04-26

**Authors:** Eileen Bigdon, Nils Alexander Steinhorst, Stephanie Weissleder, Vasyl Durchkiv, Nicole Stübiger

**Affiliations:** 1Department of Ophthalmology, University Medical Center Hamburg-Eppendorf, Martinistr. 52, 20246 Hamburg, Germany; sedgwick@gmx.net (N.A.S.); weissleder@augenaerzte-ingelheim.de (S.W.); druchkivstats@gmail.com (V.D.); n.stuebiger@uke.de (N.S.); 2Augenzentrum Schleswig-Holstein, Peter-Ox-Straße 7, 25917 Leck, Germany; 3Augenärzte Ingelheim, 55218 Ingelheim am Rhein, Germany

**Keywords:** uveitis, latent tuberculosis, posterior uveitis, retinal vasculitis, antituberculosis therapy, immunosuppression

## Abstract

Background/Aims: Controversy exists regarding 3- or 4 drug antituberculosis therapy (conventional ATT) in uveitis patients having latent tuberculosis (LTB), especially while initiating therapy with corticosteroids and/or other immunosuppressants. Methods: We performed a monocentral retrospective analysis of posterior uveitis patients with latent TB. Latent TB was diagnosed, in case of a positive QuantiFERON^®^-TB-Gold test and normal chest imaging, after ruling out other causes of infectious and noninfectious uveitis. Patients with active TB were excluded. From 2016 to 2020 we included 17 patients. Ophthalmological evaluation consisted of Best corrected visual acuity (BCVA), slit lamp examination, fundoscopy, OCT, and fluorescein- and indocyaningreen- angiography before and at months 3, 6, 12, 24, and the last follow-up after treatment. Results: Initially, all patients had active posterior uveitis with occlusive (*n* = 5 patients) and nonocclusive retinal vasculitis (*n* = 12 patients). Mean follow up was 28 ± 15 months. Therapy was started with systemic corticosteroids (mean prednisolone equivalent 71.3 mg/d) and already after 3 months it could be tapered to a mean maintenance dosage of 8.63 mg/d. Additional immunosuppressive treatment with cs- or bDMARDs was initiated in 14 patients (82%) due to recurrences of uveitis while tapering the corticosteroids <10 mg per/day or because of severe inflammation at the initial visit. While being on immunosuppression, best corrected visual acuity increased from 0.56 logMAR to 0.32 logMAR during follow-up and only three patients had one uveitis relapse, which was followed by switch of immunosuppressive treatment. As recommended, TB prophylaxis with 300 mg/d isoniazid was administered in 11 patients for at least 9 months while being on TNF-alpha-blocking agents. No patient developed active tuberculosis during immunosuppressive therapy. Conclusion: Mainly conventional ATT is strongly recommended—as monotherapy or in combination with immunosuppressives—for effective treatment in patients with uveitis due to latent TB. Although in our patient group no conventional ATT was initiated, immunosuppression alone occurred as an efficient treatment. Nevertheless, due to possible activation of TB, isoniazid prophylaxis is mandatory in latent TB patients while being on TNF-alpha blocking agents.

## 1. Introduction

Tuberculosis (TB) is a worldwide spread bacterial infectious disease. It is caused by various types of mycobacteria (Mycobacterium tuberculosis complex; it most frequently affects the lungs as pulmonary tuberculosis and might involve also any part of the eye with or without other primary foci in the body [1].

Globally, TB is one of the top 10 causes of death and the leading cause of death from a single infectious agent. Most people who develop TB are adults; there are more cases in men than in women. The 30 highest TB burden countries account for almost 90% of those who fall sick with TB each year. However, only 5–10% of the estimated 1.7 billion people infected with M. tuberculosis will develop TB disease during their lifetime.

Latent tuberculosis infection (LTBI) is characterized by the presence of immune responses to previously acquired Mycobacterium tuberculosis infection without clinical evidence of active TB.

The probability of developing TB disease is much higher among people with HIV, undernutrition, diabetes, and greater smoking and alcohol consumption [2].

TB-associated uveitis is a major cause of uveitis in tuberculosis endemic countries. The pathophysiology of TB-associated uveitis is not yet fully understood. Eye infection with Mycobacterium tuberculosis (Mtb) is a rare disease that can cause uveitis when Mtb bacteria invade the eye, causing local granulomatous inflammation [3]. Several indications suggest that autoimmune reactions against retinal antigens may be part of the pathophysiology of this special form of uveitis, which might be particularly important in the context of latent TB-associated uveitis [4]. Ocular symptoms in latent TB patients are diverse, but posterior uveitis is the most common manifestation [5].

Screening for LTBI is achieved by assessing possible Mtb exposure, immunological tests, and chest x-ray (CXR) or computer tomography (CT) to rule out active disease or to detect signs of previous TB infection. Currently, two types of immunological tests are available for the diagnosis of LTBI: tuberculin skin test (TST) and interferon gamma release assays (IGRA). The test specificity of IGRAs is almost 100%, which is much higher than in TST [6]. The QuantiFERON^®^-TB Gold test (QFT-IT; Fa. Cellestis Ltd., Carnegy, Austria) is known as the gold standard amongst the IGRAs. Alternatively, the T-SPOT^®^ test (T-SPOT, Fa. Oxford Immunotec, Oxford, UK) using the enzyme-linked immunospot (ELISpot) assay can be used [7]. Both tests are recognized by the Food and Drug Administration (FDA) and by the European Medicines Agency (EMA) as being equivalent regarding their diagnostic power for the detection of latent TB [7].

Up to now, controversy exists regarding additional 3- to 4-drug antituberculosis therapy (conventional ATT) in uveitis patients having LTBI, especially in case of treatment initiation with corticosteroids and/or other immunosuppressants. Most authors suggest that conventional ATT is necessary to treat LTBI-associated uveitis effectively and to decrease the risk of recurrence of uveitis [8,9,10,11,12]. However, in most of these studies, additional immunosuppressive treatment was necessary despite 3- to 4-drug antituberculosis therapy.

Therefore, it is most likely that uveitis in latent TB is caused by antigen recognition of isolated mycobacteria (M. tuberculosis), which triggers an excessive immune reaction, resulting in an attack against ocular tissues [5]. Based on this consideration, uveitis in latent TB seems *not* to have an infectious but an immunological origin, in contrast with uveitis in active tuberculosis, and this is the rationale for immunosuppressive treatment without conventional 3-or 4-drug ATT in uveitis patients with latent TB.

## 2. Patients and Methods

We performed a retrospective analysis of all posterior uveitis patients with LTBI from 2016 to 2020 at the Department of Ophthalmology at the University Medical Center Hamburg-Eppendorf, Germany. We included 17 patients (mean age 54 ± 16.82 years; with a range from 34 to 82 years of age; female:male = 6:11) with posterior uveitis associated with occlusive or nonocclusive retinal vasculitis, diagnosed according the SUN criteria [12,13,14,15]. Additionally, the origin of our patients was evaluated (Figure 1).

All patients with uveitis and a positive QuantiFERON^®^-TB Gold test were considered as having LTBI if they had normal chest imaging [12,13,14,15]. Other causes for uveitis were ruled out by clinical manifestation and thorough blood work including full blood count, c reactive protein (CRP), erythrocyte sedimentation rate (ESR), rheumatoid factor, antinuclear antibodies (ANAs), antineutrophil cytoplasmatic antibodies (c-ANCAs), anti-double-stranded DNA-antibodies (dsDNA-Ab), angiotensin-converting enzyme (ACE), soluble interleukin 2 receptor (s-IL-2R), Cytomegalovirus (serology test), Herpes simplex virus (serology test), Borreliosis (serology test), Syphilis (serology test), Toxoplasma (serum Enzyme-linked Immunosorbent Assay (ELISA), and Toxocara (serum ELISA). Patients with past or present active tuberculosis were excluded.

The primary question was whether immunosuppressive treatment without conventional ATT is effective in LTBI patients with posterior uveitis. In addition, we ruled out if activation of TB occurred under immunosuppression, when only isoniazid prophylaxis (300 mg/d) was added, in case of initiating TNF-alpha-blocking therapy.

Secondary endpoints were the number of uveitis recurrences while tapering the systemic corticosteroids or immunosuppressives; the dosage of systemic corticosteroids; the change in best-corrected visual acuity (BCVA), the number of patients in whom additional immunosuppression was required, the number and severity of adverse events (AEs), and the number of patients requiring dose reduction or discontinuation of systemic corticosteroids or immunosuppression due to adverse events.

### 2.1. Treatment

In collaboration with the Department of Infectious Diseases at the University Medical Center Hamburg-Eppendorf, we developed an immunosuppressive regime without primary 3- to 4-drug-antituberculostatic therapy based on the rationale that uveitis in latent TB is mainly caused by an immunological reaction.

At first, all patients received oral prednisolone equivalent at an initial dose of usually 1–2 mg/kg bodyweight at the primary visit, followed by weekly tapering to a maintenance dosage of 7.5 mg/d, respectively, 5.0 mg/d (Figure 2). In case of relapse while patients were on systemic corticosteroids at a dosage of ≥10 mg prednisolone equivalent immunosuppression with conventional Disease-Modifying Drugs (csDMARDs), methotrexate (MTX) or azatioprine (Aza) were initiated. In case of further relapse after 3 months of treatment with csDMARDs, the biological DMARD (bDMARD) adalimumab was applied additionally (because the TNF-alpha blocking agent adalimumab (ADA) is approved for posterior uveitis in Germany). In uveitis patients with severe ocular inflammation, csDMARDs or adalimumab were initiated immediately. In all patients who are on adalimumab and having no contraindications against MTX, low-dose methothrexate (5 mg weekly) was added to avoid the formation of antidrug-antibodies against adalimumab; in addition, for every patient who was on ADA treatment and, in some cases with other immunosuppressants, isoniazid (INH)-prophylaxis (300 mg/d) was added, because the administration of TNF-alpha blockers and also other csDMARDs increase the risk of TB, especially by reactivating LTBI. The guidelines of the German society of rheumatology recommends prophylaxis with INH (300 mg/d) for at least 9 months starting 4 weeks prior to the initiation of adalimumab if possible [16,17].

Efficacy of treatment was assessed by standard ophthalmic examination techniques, including measurement of visual acuity (VA in logMAR) in both eyes (VA of hand motion and counting fingers was quantified with the Freiburg Visual Acuity Test [18]), slit-lamp examination, indirect binocular ophthalmoscopy, fluorescein and/or indocyanin green angiography, and Optical Coherence Tomography (OCT) imaging to determine disease severity. The examinations were performed before and in months 3, 6, 12, 18, and 24 after treatment initiation; in patients with a follow up of more than 24 months, a last follow-up visit was added.

### 2.2. Statistical Analysis

In this study, only 4 patients had a full set of data, so our main problem for statistical analysis was the missing data for follow up visits, because traditional statistical methods would discard all patients with at least one missing value. To efficiently deal with this problem, we analyzed the changes in best-corrected visual acuity (BCVA in logMar) and in systemic corticosteroid treatment via the mixed regression model by Pinheiro and Bates (2006). This method allows us to use all data available, and not only nonmissing time trajectories, and to perform multivariate analysis with interaction terms. The overall effect of time or interaction between a covariate and time was assessed with the ANOVA method. In cases where it was considered necessary, we repeated the estimation using robust regression method of Koller (2016) [19]. The pairwise comparisons were performed, and *p* values were adjusted via Tukey adjustment method. A *p*-value of ≤0.05 was considered statistically significant. 

All analysis were performed with R Core Team (2019) [20].

## 3. Results

### 3.1. Ocular Involvement

Before treatment initiation, all 17 LTBI patients were diagnosed with active posterior uveitis (chorioretinitis) associated with occlusive retinal vasculitis (*n* = 5 patients) or nonocclusive retinal vasculitis (*n* = 12 patients). Mean follow-up was 28.0 ± 15.0 months. Altogether, 30 eyes were affected, in 13 patients it was both eyes, while 4 patients had only unilateral involvement.

### 3.2. Treatment

#### 3.2.1. Corticosteroids

Treatment was initiated with systemic corticosteroids (1–2 mg/kg bodyweight prednisolone equivalent orally or intravenously) in 16 patients (94%). One patient did not tolerate the systemic steroids and received monotherapy with adalimumab instead. During follow-up, in only 3 patients (19%) was corticosteroid monotherapy was effective.

Figure 3 demonstrates the mean dosage of corticosteroids at treatment initiation and the mean decrease during a follow-up time of two years.

The mean initial dose of systemically applied corticosteroids in our patients was 71.03 mg/d (standard (std.) error 1.695). At month 3, there was a significant decrease to a mean dosage of 8.63 mg/d (std. error 1.85, *p* < 0.001) and a final mean dosage of 2.83 mg/d at month 24.

#### 3.2.2. Immunosuppression (cs- and bDMARDs)

During follow-up, additional immunosuppressive treatment with cs- or bDMARDs was initiated in 14 patients (82%) (Figure 4, Table 1) either due to recurrences of uveitis while tapering the systemic corticosteroids ≥10 mg/d (3 patients) or because of severe inflammation at the initial visit (11 patients). During the course of 24 months, there were 8 patients in whom treatment was changed at least once (pat.no. 1, 2, 7, 8, 10, 12, 13, 14). In 3 patients (18%), treatment had to be switched twice (pat.no. 7, 8, 13) (Figure 4).

In 10 patients (59%), immunosuppression in addition to corticosteroid treatment has to be started at the primary visit: in four patients (pat.no. 2, 3, 6, 10), azathioprine in a dosage of 150 mg/d orally; in one patient (pat.no. 11), adalimumab 40 mg every 2 weeks subcutaneously; and in four patients (pat. no. 1, 12, 15, 16), methotrexate (MTX) 10–25 mg/week subcutaneously. One patient (pat.no. 17) was already on cyclosporine A (CSA 3 mg/kg bodyweight daily) because of an earlier corneal transplant. Another patient (pat.no. 14) did not tolerate the systemic steroids and received monotherapy with adalimumab instead. In six patients (pat.no. 4, 5, 7, 8, 9, 13), corticosteroid monotherapy was effective—at least for the first 3 months.

At the 3-month visit, in three patients who had a uveitis relapse while tapering the corticosteroid monotherapy, additional therapy with azathioprine (pat.no. 8, 13) or adalimumab in combination with low-dose MTX (5 mg/week) (pat.no. 7) was initiated.

After a follow-up of 6 months, treatment was changed from azathioprine to MTX due to side effects (elevated liver enzymes) in patient no. 8.

At month 12, one patient’s (pat.no. 12) therapy was switched from MTX to azathioprine (due to severe hair loss) and in two patients (pat.no. 13, 14) from MTX to adalimumab in combination with low-dose MTX (5 mg/week); in patient 13, this had to be done due to a uveitis relapse, and in patient 14, low-dose MTX was added to prevent the formation of antibodies against adalimumab.

At month 18, another patient (pat.no. 7) received additional low-dose MTX to his ADA treatment for reducing the risk of formation of antidrug-antibodies against ADA.

At month 24, patient no. 4 stopped low-dose corticosteroid treatment on his own because of personal problems due to weight gain. Patient no.1 stopped MTX treatment—also on his own—due to gastrointestinal problems; then, he restarted it upon our request and reported no further problems.

At the last visit, after 36 months, in patient no. 2, azathioprine treatment was switched to adalimumab in combination with low-dose MTX (due to persisting macular edema). After 48 months, pat. no. 1 was able to stop all immunosuppressive treatments due to complete remission. 

One patient (pat.no. 16) was lost for follow-up after 3 months, two patients (pat.no. 9, 17) were lost for follow-up after 6 months, and one patient (pat.no. 15) after 12 months.

Interestingly, all patients who had occlusive retinal vasculitis (pat.no. 3, 12, 13, 16, 17) and only 75% of patients with nonocclusive retinal vasculitis (pat.no. 1, 2, 6, 7, 8, 10, 11, 14, 15) required additional immunosuppressants. 

#### 3.2.3. Efficacy of Immunosuppression: Relapses of Uveitis/Steroid-Sparing Effect

During corticosteroid monotherapy initiation of an additional treatment with csDMARDs was necessary in three patients (pat.no. 7, 8, 13) at month 3 because of a uveitis relapse while tapering the corticosteroids to ≤10 mg prednisolone equivalent daily. At month 18, one patient (pat.no. 13) needed to be switched to ADA (with low-dose MTX) because of recurrence of uveitis during azathioprine treatment for 9 months. At the last visit, at month 36, in patient no. 2, azathioprine was switched to adalimumab in combination with low-dose MTX due to persisting macular edema.

Pat.no. 1 had recurrence of his uveitis due to stopping immunosuppression on his own after 9 months, and on the patient’s request he received high-dose corticosteroids in addition to MTX. After 18 months, he was able to stop all immunosuppressants.

Altogether, during corticosteroid monotherapy, there were three uveitis relapses; during treatment with csDMARDs, two patients’ treatments had to be switched due to recurrence of uveitis and in one patient due to chronic persisting CME. No relapse occurred in all patients who were on ADA.

Another sign for the efficacy of immunosuppression is its steroid-sparing effect; at treatment initiation, patients had a mean dosage of systemic corticosteroids of 71.03 mg/d and, at the end of follow-up, the dosage was decreased significantly by 68.27 mg to a mean of 2.83 mg/d (Figure 2 and Figure 4).

#### 3.2.4. Isoniazid Prophylaxis

As prophylaxis for TB reactivation, we administered isoniazid (INH) 300 mg/d in 11 patients in addition to immunosuppression for at least 9 months. This INH prophylaxis was performed in all patients receiving adalimumab (pat. no. 7, 11, 13, 14), in one patient receiving cyclosporine A (pat.no. 17), in 3 patients who were on azathioprine (pat.no. 2, 3, 6), and in 3 patients who received corticosteroid monotherapy (4, 5, 9). None of these patients developed active tuberculosis.

#### 3.2.5. Side Effects of Systemic Therapy

One patient reported a single episode of pressure in the throat after taking azathioprine. The patient continued the medication and the pressure subsided completely (pat.no. 6).

Due to elevated liver enzymes, azathioprine was switched to MTX (pat.no. 8, 10) and to adalimumab in two patients (pat.no. 2, 13). During MTX treatment, one patient developed severe hair loss; therefore, the medication was changed to azathioprine (pat.no. 12). Another patient (pat.no. 1) stopped MTX on his own due to gastrointestinal problems and continued it later without further problems. One patient reported reactivation of dermatological herpes simplex virus (HSV) infection under adalimumab. He was treated with systemic aciclovir and immunosuppression could be continued with no further HSV recurrence (pat.no. 7).

#### 3.2.6. Intravitreal Treatment

Seven patients received intravitreal injections. 

Three patients (pat.no. 2, 7, 13) were injected with Ozurdex^®^ intravitreally (ivi), one patient with Iluvien (pat.no. 7), and two patients (pat.no. 2, 17) received triamcinolone as a parabulbar injection due to macular edema.

Three patients (pat.no. 1, 13, 16) with occlusive retinal vasculitis received Bevacizumab and one patient (pat.no. 8) Ranibizumab ivi due to retinal neovascularizations (Table 1).

### 3.3. Visual Acuity 

The mean best-corrected visual acuity (BCVA) of all patients at baseline was 0.56 logMar (Logarithm of the Minimum Angle of Resolution) The standard deviation between the eyes was 0.51 (LogMar) and within the eyes 0.29 (LogMar). The overall effect of time was significant (*p* = 0.045). Visual acuity (VA) at month 3 and at month 18 increased significantly in comparison to VA before treatment initiation. Other pairwise comparisons during treatment at months 6, 12, and 24 were not statistically significant (*p* > 0.05) (Figure 5).

At the end of the observation period, the mean BCVA of all patients was 0.32 logMar.

#### 3.3.1. Visual Acuity in Patients with Occlusive Retinal Vasculitis versus Patients with Nonocclusive Retinal Vasculitis

Baseline BCVA in the nonocclusion group was 0.435 (Std. error. 0.18) and in the occlusion group, 0.8, *p* = 0.103. The BCVA in the nonocclusion group was 0.18 and in the occlusion group, 0.9. The differences between the groups were significant using robust approach (Figure 6). Due to the ischemic nature of occlusive retinal vasculitis, the VA prognosis is known to be worse compared with patients with nonocclusive retinal vasculitis. In our group, five patients had occlusive retinal vasculitis (pat.no. 3, 12, 13, 16, 17) and 12 patients had nonocclusive retinal changes. The cases with occlusive retinal vasculitis (occlusion group) have worse vision trajectory. The overall difference was significant, F (1,28) = 7.96, *p* = 0.012. They had negative changes when compared with cases with nonocclusive retinal vasculitis (nonocclusion group), interaction F (5117) = 3.22, *p* = 0.0092. All intervals during treatment were significantly different to baseline in cases with no occlusion, *p* < 0.05. No significant change was observed in the occlusion group, *p* > 0.05.

#### 3.3.2. Uveitic Macular Edema

Ten patients (pat.no. 1, 2, 4, 7–9, 11–13, 17) were diagnosed with cystoid macular edema (CME) (60% of patients with occlusive retinal vasculitis, 67% of patients with nonocclusive retinal vasculitis). However, in one patient (pat.no. 1), the edema was caused by diabetic retinopathy most likely. Except in three patients (pat.no. 2, 13, 17), CME resolved completely during immunosuppressive treatment. In two (pat.no. 13, 17) of these three patients, CME disappeared after Ozurdex^®^ ivi (pat.no. 13), Iluvien^®^ (pat.no. 7), or triamcinolone parabulbar (pat.no 17) (Figure 7A–C). Only patient no. 2 disclosed persistent macular edema, even after three injections of Ozurdex^®^. Therefore, therapy was changed from azathioprine to adalimumab with low-dose MTX at month 24.

#### 3.3.3. Other Uveitis-Associated Complications

In the patient group with occlusive retinal vasculitis (Figure 8A,C,D and Figure 9A,B), three patients (pat.no. 13, 16, 17) and only one patient with nonocclusive retinal changes (pat.no. 6) developed vitreous hemorrhage due to retinal neovascularizations and/or active retinal vasculitis. In all of these, patients’ immunosuppressants were escalated and retinal laser coagulation was performed (Figure 8A–D). Additionally Bevacizumab (Avastin^®^) (pat.no. 16, 13) and Ranibizumab (Lucentis^®^) (pat.no. 8) were injected intravitreally.

#### 3.3.4. Other Complications

Seven patients (pat.no. 2, 9, 10, 11, 12, 13, 16) showed retinal scarring. In one patient (pat.no. 2), the scarring was caused by vitrectomy, which had been performed even before treatment initiation, and only one patient (pat.no. 17) developed secondary glaucoma during the course of disease, which remained stable during locally applied brimonidine therapy.

## 4. Discussion

### 4.1. Study Design

Different study groups [9,10,11,12] have concluded that 4-drug ATT is necessary to reduce the recurrences of uveitis. Bansal et al. [12] performed a retrospective interventional case series comparing two groups. Group 1 received 4-drug ATT and group 2 received corticosteroid monotherapy only. Their data demonstrated that uveitis relapse was statistically significantly less likely in the first group. They therefore concluded that 4-drug ATT reduced the risk of recurrence compared with corticosteroid monotherapy. However, they did not apply any additional immunosuppressive treatment in case of uveitis relapse while tapering the corticosteroids. In contrast, in 82% of our patients (14 patients), additional immunosuppression was necessary.

Ang et al. compared uveitis patients who received only ATT for 9 months to patients who discontinued the medication before [10,11]. However, they did not specify the exact medications that were given to the patients, but no additional immunosuppressive therapy was initiated. Only Tomkins-Netzer et al. compared LTBI uveitis patients who received ATT to patients who were treated with corticosteroids or, in 9% of these patients, with corticosteroids in association with immunosuppressants. However, this publication did not differentiate between patients receiving corticosteroid monotherapy or treatment combinations. In addition, the authors did not mention which immunosuppressants were administered in their patients. Altogether, uveitis relapses were statistically significantly less likely in patients treated with ATT (Tomkins-Netzer) [9].

In our study, only patients in whom systemic corticosteroids were not sufficient or severe inflammation was already present at the primary visit received additional medication. This potentially avoids serious side effects in 20% of the patients because serious adverse reactions to antituberculosis drugs are common [21] and immunosuppressants usually have less-severe side effects and are better tolerated than regular ATT. In addition, ATT is commonly recommended for 3–9 months in patients with LTBI [22,23]. There are no official recommendations for treatment duration of ATT in uveitis patients with LTBI. So, in real life, ATT application for 9 months and more seems to be the consensus in such uveitis patients. Serious side effects of ATT are observed in about 7% of patients, including hepatotoxicity (6%) and skin rash (0.6%). Gastrointestinal side effects, ocular toxicity, angioedema, and other side effects are comparatively rare with less than 0.4%. However, early diagnosis of ocular toxicity, which primarily affects the optic nerve, is crucial to prevent the potential loss of function [24,25,26]. Isoniazid 600–750 mg/d can cause neurological toxicity and hepatitis. Although much less frequent, hepatotoxicity can lead to liver transplantation or even to death [26,27]. The risk of hepatotoxicity due to isoniazid treatment in patients older than 65 years may be increased 3- up to 5-fold [28]. Rifampicin may cause gastrointestinal reactions or thrombocytopenic purpura and, rarely, shortness of breath, shock, acute hemolytic anemia, and acute renal failure. Pyrazinamide can lead to arthralgia and hepatitis. Ethambutol may cause a dose-related retrobulbar neuritis.

### 4.2. Methods of TB Testing

Bansal et al. and Ang et al. [10,11,12] used the tuberculin skin test (TST) with purified protein derivation (PPD) or recombinant purified protein derivates. Intradermal injection leads to delayed-type hypersensitivity between 48 and 72 h. The test is susceptible to placement errors, reading errors, false negatives in anergic patients, false positives in Bacille Calmette-Guérin (BCG)-vaccinated patients, the booster phenomenon (where repeated testing induces a positive result), and noncompliance, as it requires two visits to the physician. The test has a low specificity and sensitivity (reported sensitivity of 71% (95% CI (confidence interval), 65–74) and specificity of 66% (95% CI, 46–86) [12].

However, the gold standard in TB testing is the QuantiFERON^®^-TB Gold test. After presentation of mycobacteria-specific antigens (ESAT-6, CFP-10, TB 7.7) the interferon production of specific T-lymphocytes of the patient is measured. The antigens used are largely specific to the Mycobacterium tuberculosis complex, but not to the BCG vaccine strain. Thus, an earlier vaccination does not lead to a positive test result. The QuantiFERON^®^-TB Gold test has a specificity of 99% [29]. Therefore, TB testing, which we used in this study design, is clearly superior. Alternatively, the T-SPOT.TB test (T-Spot) can be used. It uses the Enzyme-linked immunospot (ELISPOT) methodology to similarly count the T cells that have been sensitized by TB infection. Sensitivity and specificity vary between different studies. The sensitivity of QuantiFERON^®^ of the new generation is higher than that of the older generation (94% vs. 81%), thus reaching the sensitivity of T-SPOT^®^.TB [7].

### 4.3. Recommended Therapy of Latent Tuberculosis

The presumed pathogenesis of uveitis related to latent TB infection consists of inflammation caused by presence of low numbers of bacteria within the eye with or without superimposed immune reaction to mycobacterial or ocular antigens [5]. This suggests that immunosuppression might be the superior therapeutic strategy in that case. This thesis is also supported by the WHO guidelines for latent tuberculosis. 

The main principle in guiding testing and treatment for LTBI is that the benefit outweighs the risk to the individual, while the WHO guidelines strongly recommend systematic testing and treatment of LTBI for people living with HIV, adult and child contacts of pulmonary TB cases, patients initiating antitumor necrosis factor inhibition treatment, patients receiving dialysis, patients preparing for organ or hematological transplantation, and patients with silicosis. Treatment and testing in any other group in countries with low incidence of TB (<10 cases per 100,000 population per year) is not generally recommended. In 2019, 54 countries had a low incidence of TB, mostly in the American and European region, plus a few countries in the Eastern Mediterranean and Western Pacific regions [2].

In contrast to tuberculosis requiring treatment, the germ populations in LTBI are probably very small. This assumption is based on the results of animal studies; however, it is unclear to what extent the bacterial burden of the individual can be reduced in LTBI, as there are no suitable methods to accurately quantify the intracellular pathogens. The probability of spontaneous resistance mutation of M. tuberculosis in the presence of LTBI in humans is considered very low. Therefore, monotherapy of LTBI is sufficiently safe (Exception: assumed infection by INH- or RMP-resistant pathogens) [30].

The proposed treatment regimens for LTBI include the following: 6- or 9-months isoniazid; or 3-months rifampicine (RIF) plus isoniazid; or 3–4 months isoniazid plus rifampicin; or 3–4 months rifampicine alone [21]. Treatment dosage varied among the trials: INH 300 mg or 600 mg for daily regimens; 600 mg or 900 mg for twice weekly regimens; RIF 450 mg or 600 mg. The risk of significant adverse events during LTBI treatment is approximately 3% [31] and 8% in ATT [32].

### 4.4. Efficacy of Preventive Therapy for LTBI

A sufficient number of controlled trials are available to assess the efficacy of preventive therapy with INH, and each has been analyzed in a Cochrane meta-analysis for both HIV-positive [32] and HIV-negative individuals [33]. Treatment dosage varied among the trials: INH 300 mg or 600 mg for daily regimens; 600 mg or 900 mg for twice weekly regimens [32,33].

The number needed to treat (NNT) chemo-preventively to prevent tuberculosis ranges from 30 to 89 in immunocompetent individuals and from 14 to 80 in immunocompromised individuals [18]. It is possible that therapeutic regimens based on Rifamycine derivatives are even more effective than INH monotherapy for the preventive treatment of LTBI [18].

As the recommendations state, preventive LTBI therapy is necessary when initiating treatment with antitumor necrosis factor [17] and the SAFEBIO study [18] demonstrated that even csDMARDs and high-dose systemic corticosteroids can reactivate TBC; all patients with adalimumab and some patients with corticosteroids and/or csDMARDs in our study received 300 mg isoniazid daily for at least 6 months. In total, 11 patients were medicated with preventive INH therapy in addition to immunosuppressive treatment and none of our study patients disclosed TB reactivation.

### 4.5. Efficacy of Immunosuppressive Therapy in Uveitis Patients with LTBI

The mean best-corrected visual acuity (BCVA) at baseline was 0.56 ± 0.51 logMar and increased significantly to 0.36 logMar at the last follow-up. The mean initial dose of systemically applied corticosteroids in our patients was 71.03 mg/d (standard (std.) error 1.695) and, at month 3, there was a significant decrease to a dose of 8.63 mg/d (std. error 1.85, *p* < 0.001) and to a dose of 2.83 mg/d at month 24.

In our study, three patients had only one uveitis relapse while they were on corticosteroid monotherapy, during treatment with csDMARDs. In only two patients did treatment have to be switched due to recurrence of uveitis; in patients who were on ADA, no relapse occurred. Altogether, five relapses occurred during immunosuppressive treatment in 17 patients during a mean follow-up at 28 months; thus, 71% of our patients had no relapse. Ang et al. [7] only included patients that did not have a recurrence of uveitis 1 year after 4-drug ATT therapy, which was administered for at least 6 months. In their study about the duration of ATT, they achieved that only 14.4% of patients who applied corticosteroid monotherapy had no recurrence of their uveitis while 24% of patients who received 4-drug ATT did not have a recurrence after 6 months after completing the therapy [8]. Bansal et al. found that patients treated with 4-drug ATT and systemic corticosteroids had a 16% chance of a uveitis relapse while patients treated with corticosteroids alone had a likelihood of 47% [12]. The patient groups of Tomkins-Netzer et al. [9] disclosed a 30% rate of uveitis relapse in the ATT-treated group, and in 48% of patients who received corticosteroid as monotherapy or in combination with immunosuppressants.

## 5. Conclusions

In our study, 18% of patients improved with corticosteroid monotherapy alone, sparing those patients additional side effects caused by stronger immunosuppression. A total 82% of patients received immunosuppressive therapy with cs- or bDMARDs in addition to systemic corticosteroids, when corticosteroid monotherapy was not sufficient. Eleven patients (65%) received additional isoniazid prophylaxis with 300 mg/d, while six patients did not receive any antituberculostatic therapy. Our patients had significant improvement of BCVA. Of our patients, 71% had no uveitis relapse; in only five patients, a recurrence of uveitis occurred during the follow-up of 28 months, which made a treatment switch necessary. The mean dosage of systemic corticosteroids could be significantly reduced to a mean of 2,76 mg/d at the end of follow-up. Therapy with corticosteroids and immunosuppressives with additional INH prophylaxis proved to be a safe treatment option in patients with uveitis in latent tuberculosis. No patient developed active tuberculosis during the follow-up period.

Our data demonstrate that additional 3- to 4-drug ATT seems not to be necessary for efficient uveitis treatment in patients with latent tuberculosis. Especially, when comparing the recurrence rates with ATT-treatment studies [8,9,10,11,12], we could achieve the lowest rate with 71% of relapse-free patients. This study further supports the thesis that uveitis in latent tuberculosis is rather caused by immunoreactions to mycobacterial or ocular antigens in the presence of only low numbers of tuberculosis bacteria and not by infection.

It is most important to keep in mind that uveitis patients with LTBI must have INH prophylaxis with 300 mg/d for at least 9 months in case of initiating TNF-alpha-inhibiting treatment [14,18]. Whether INH prophylaxis should be initiated in every patient diagnosed with LTBI is also controversial, because in low TB burden countries, patients without risk factors to develop active TB, such as an HIV infection, are not likely to develop active TB. Most probably, INH prophylaxis is not necessary in every patient with uveitis posterior due to latent TB. Further, in our study, six LTBI patients did not receive INH prophylaxis and none of them developed active TB.

Interestingly, the WHO guidelines on the therapy for LTBI in countries with a low prevalence of TB only recommend Isoniazid or Rifampicin monotherapy or a combination of both, instead of a 3- or 4-drug ATT [18]. However, until today, this therapy regime has not been widely recognized by ophthalmologists, maybe due to the fear of TB activation. So, since Isoniazid monotherapy (300–750 mg/d) [30] for 6–9 months is sufficient treatment for LTBI itself, the administration of INH might also have an impact on the uveitis.

Since this cohort is rather small, further investigations are needed to support this thesis.

## Figures and Tables

**Figure 1 jcm-11-02419-f001:**
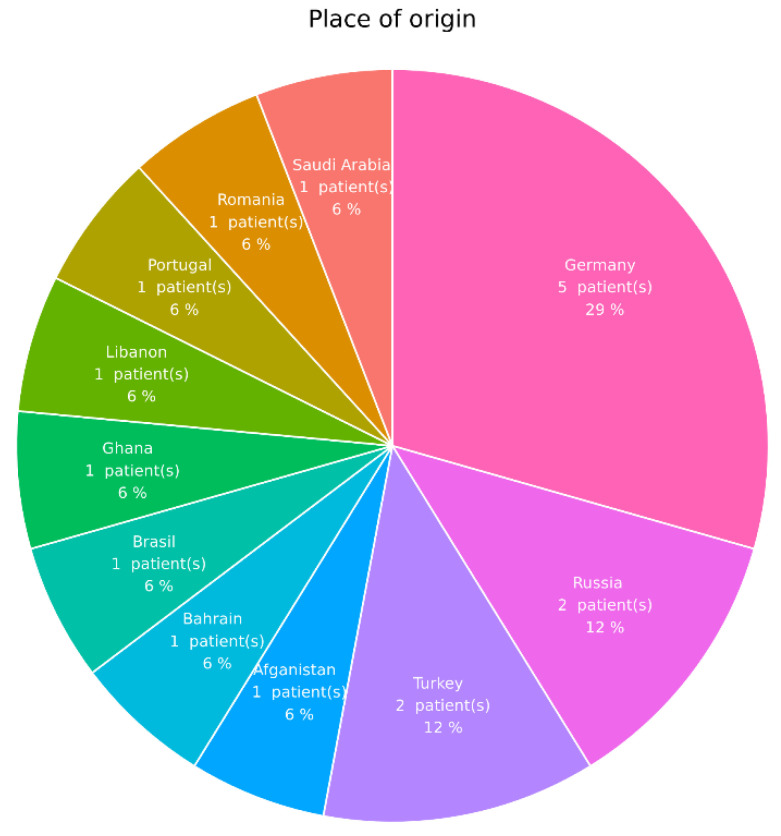
Shows the distribution of countries where the patients originated: Germany, 29.4% (*n* = 5); Russia, 11.8% (*n* = 2); Turkey, 11.8% (*n* = 2); Afghanistan, 5.88% (*n* = 1); Bahrain, 5.88% (*n* = 1); Brazil, 5.88% (*n* = 1); Ghana, 5.88% (*n* = 1); Lebanon, 5.88% (*n* = 1); Portugal, 5.88% (*n* = 1); Romania, 5.88% (*n* = 1); Saudi Arabia, 5.88% (*n* = 1).

**Figure 2 jcm-11-02419-f002:**
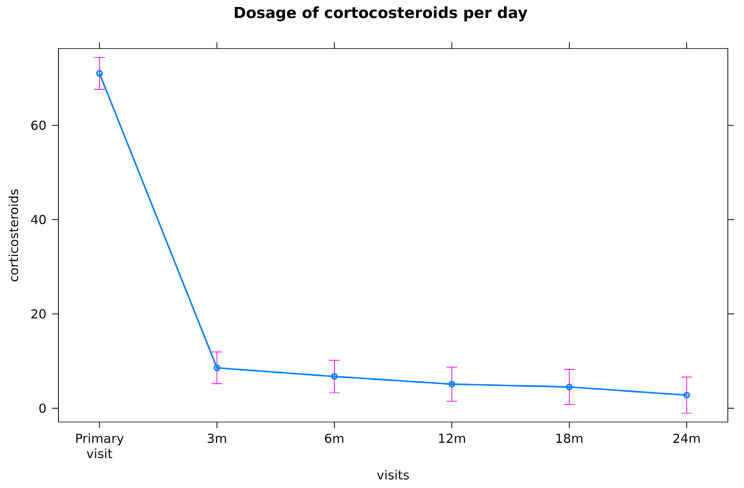
Shows the patients’ daily demand of systemic corticosteroids (mean dosage in mg/d and their 95% confidence intervals (CI)) at the primary visit FUO (71.03 mg/d) and the mean decrease at FU1 (3-month visit: 8.63 mg/d), FU2 (6-month visit: 6.72 mg/d), FU3 (12-month visit: 5.09 mg/d), FU4 (18-month visit: 4.5 mg/d), and FU5 (24-month visit:2.83 mg/d); m = months.

**Figure 3 jcm-11-02419-f003:**
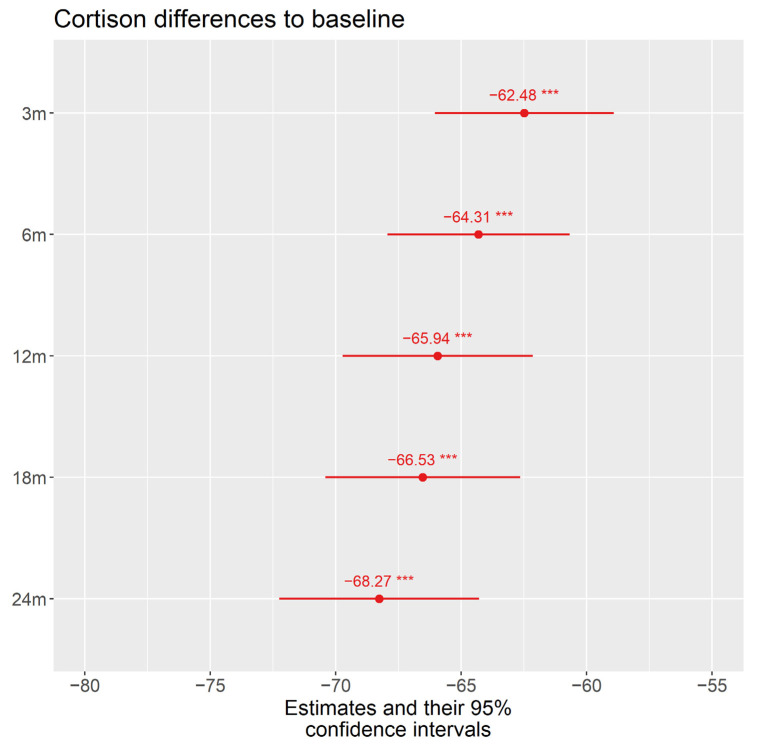
Shows the patients’ daily demand of systemic corticosteroids (mean dosage in mg/day and their 95% confidence intervals) in contrast with the initial visit FUO (71.03 mg/d) and the mean decrease at FU1 (month 3 visit: −62.48 mg/d), FU2 (month 6 visit: −64.31 mg/d), FU3 (month 12 visit: −65.94 mg/d), FU4 month 18 visit: −66.53 mg/d), and FU5 (month 24 visit: −68.27 mg/d); m = months. *p* value *** 0.001.

**Figure 4 jcm-11-02419-f004:**
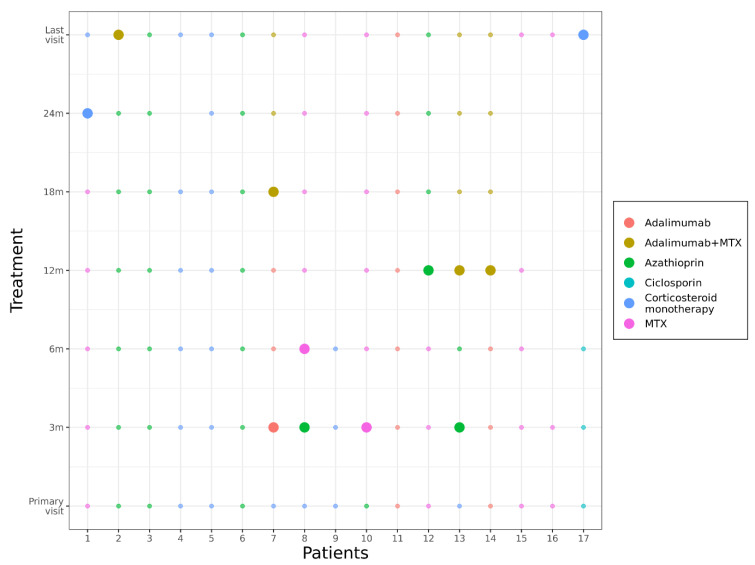
Shows the different treatment options and the switch of therapy of each patient during the follow up in months (m). Each color represents a different therapy; the larger dots indicate a switch of immunosuppression. Patient no. 9, 15, 16, 17 did not have complete follow-up.

**Figure 5 jcm-11-02419-f005:**
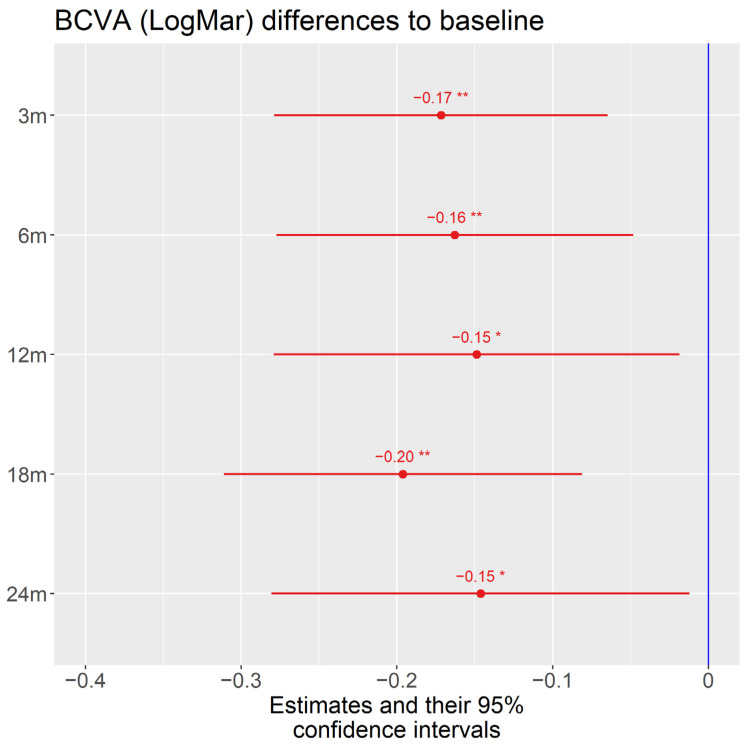
The estimates and their 95% confidence intervals in BCVA (logMar) from baseline. A total 157 BCVA data were used for the analysis. The overall effect of time was significant, F (5122) = 2.35, *p* = 0.045. Visual acuity at FU1 (month 3) and FU4 (month 18) was significantly different from initial visual acuity when performing pairwise comparisons between all time intervals and adjusting *p*-values. Other pairwise comparisons were not statistically significant *p* > 0.05. The data were analyzed via mixed regression modeling @pinheiro2006mixed. (Note: Some severe cases with bad vision caused decreased mean values for visual acuity, which led to large standard errors and, thus, less statistical significance. For this reason we applied robust mixed regression as well to confirm the findings. In this model, all mean values during treatment were statistically significantly different from baseline but the magnitude of change was somewhat less than estimated with the nonrobust method. All changes during treatment were estimated to be of around 1 Snellen line (−0.1 LogMar change)). FU1: 3-month visit (3 m), FU2: 6-month visit, (6 m) FU3: 12-month visit 12 m), FU4: 18-month visit (18 m), FU5: 24-month visit (24 m), BCVA: best-corrected visual acuity, logMAR: Logarithm of the Minimum Angle of Resolution. *p* value * 0.05 ** 0.01.

**Figure 6 jcm-11-02419-f006:**
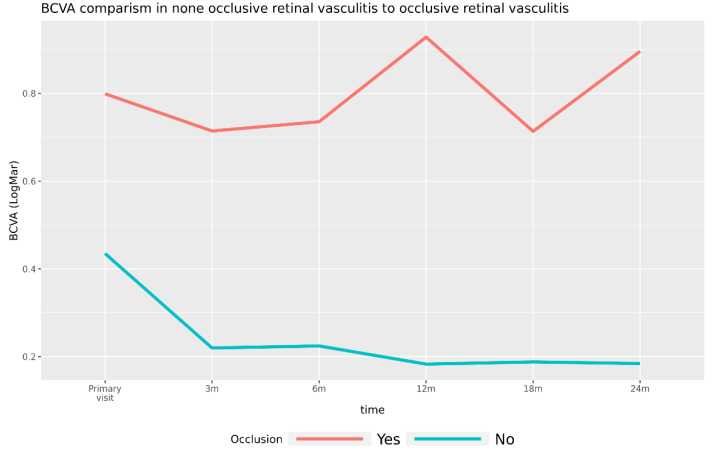
Shows the BCVA in patients with nonocclusive retinal vasculitis (green line) compared patients with occlusive retinal vasculitis (red line). The cases with occlusion have worse vision trajectory. The overall difference was significant, F (1,28) = 7.96, *p* = 0.012, where cases with occlusion had overall worse vision. Furthermore, they have negative changes compared with cases without occlusion (interaction, *p* = 0.0092). The data were analyzed with the ANOVA method. All postoperative intervals were significantly different to baseline in cases with no occlusion, *p* < 0.05. No significant change was observed in the group with occlusion, *p* > 0.05. Baseline BCVA in the nonocclusion group was 0.435 (Std. error. 0.18) and in the occlusion group, 0.8, *p* = 0.103. The differences between the groups at postoperative visits were all significant, with occlusion group having worse vision, *p* < 0.05. (Note: Due to some severe cases, we also repeated the analysis with robust estimation. The trend of the occlusion group having worse vision was confirmed by this method as well. The difference at baseline, which lacked statistical significance, was found significant using robust approach).

**Figure 7 jcm-11-02419-f007:**
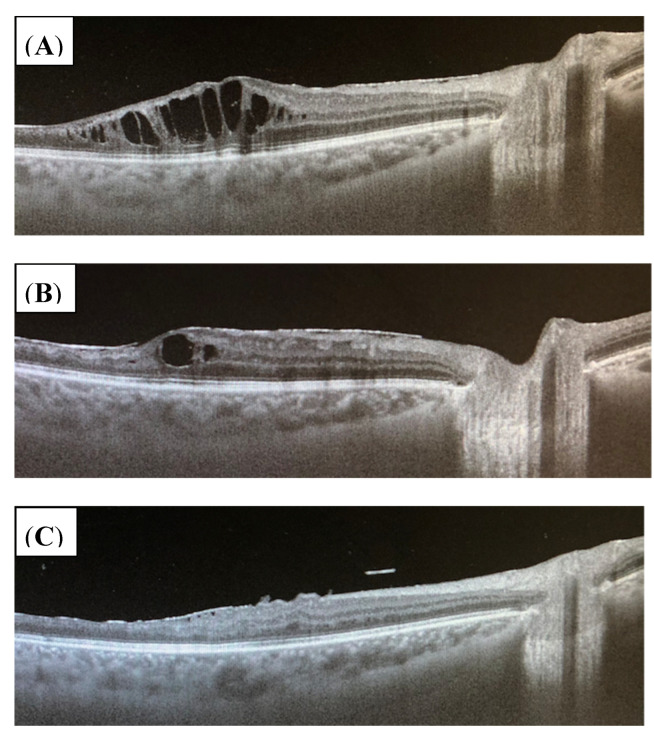
(**A**–**C**) OCT of the right eye (OD) of patient no. 13 showing cystoid macular edema before treatment (**A**), after 4 months while being on 30 mg/d corticosteroids and azathioprine 150 mg/d (**B**), and at the last follow-up after 41 months during adalimumab 40 mg every 2 weeks sc, methotrexate 10 mg/weekly, and 1 mg/d prednisolone (**C**). Macular edema regressed completely over time.

**Figure 8 jcm-11-02419-f008:**
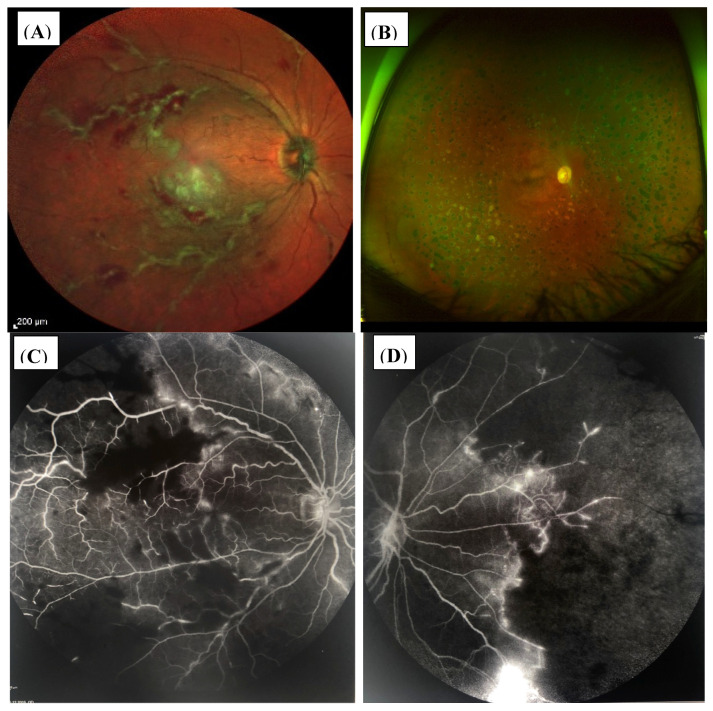
(**A**–**D**). Fundus OD of patient no. 13. (**A**) Active occlusive retinal vasculitis before initiating anti-inflammatory therapy. (**B**) Demonstrates the fundus appearance at month 24 after intravitreal Dexamethason and Bevacizumab injections and panretinal laser coagulation, due to CME and retinal neovascularizations resulting from massive retinal ischemia. BCVA was 0.1 logMar and the patient received immunosuppressive treatment with low-dose corticosteroids (7.5 mg/d), adalimumab (40 mg sc every other week), and low-dose MTX (5 mg weekly). (**C**,**D**) Fluoresceine angiography (FAG) of OD at the initial visit showing severe occlusive retinal vasculitis at the central posterior pole (**C**) and massive ischemia with nonperfusion areas and retinal neovascularizations at the posterior segment (**D**).

**Figure 9 jcm-11-02419-f009:**
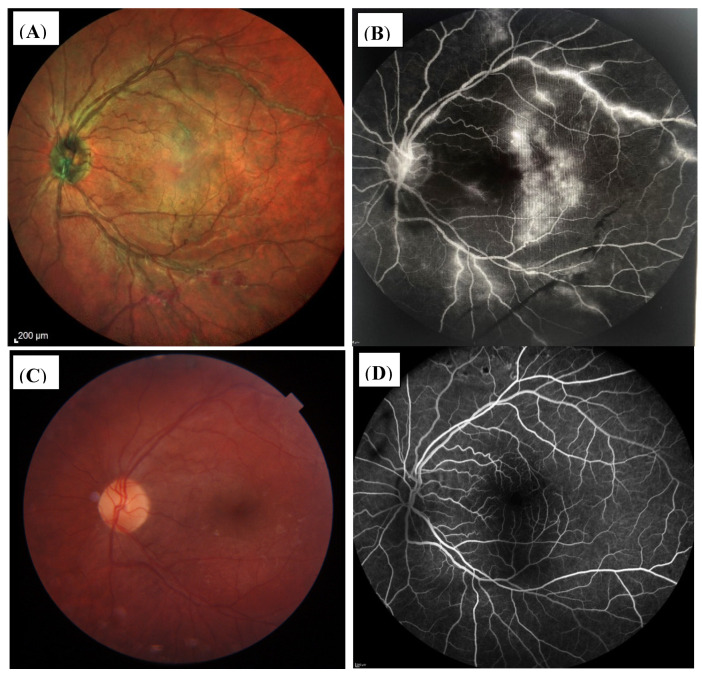
(**A**–**D**) Shows the retina of the left eye (OS) of patient 13 before and after therapy. (**A**) (fundus picture) and (**B**) (fluoresceine angiography (FAG) show occlusive retinal vasculitis. (**C**) (fundus picture) and (**D**) (FAG) display the OS after treatment, showing a normal fundus appearance and FAG. The BCVA was 0.0 logMar.

**Table 1 jcm-11-02419-t001:** Shows the age, duration of therapy, BCVA before and after treatment, treatment, adverse events, intravitreal therapy, and if macula edema was present. Cort = corticosteroids, MTX = Methotrexate, AZA = Azathioprin, ADA = Adalimumab. OD: oculus dexter, right eye; OS: oculus sinister, left eye.

Patient	Age (Years)	Sex	Quantiferon Test	Follow Up (Month)	Disease Onset	Type of Retinal Vasculitis	BCVA OD at Begin of Follow Up LogMar	BCVA OD at Begin of Follow Up LogMar	BCVA OD at Last Follow Up LogMar	BCVA OS at Last Follow Up LogMar	CME	CME End of Follow Up	Intravitreal Therapy	Remission State at Last Follow Up	Systemic Therapy (cs-/bDMARDs/Corticosteroids) at Treatment Initiation	Adverse Events	Therapy Change (or Dropout (Reasons)	Systemic Corticosteroids (mg/d) at Last Follow Up	Systemic Therapy at Last Follow Up
1	54	m	positive	33	01/18	Occlusive OD + OS	0.4	1.3	0.2	0.2	yes OD < OS	no	Bevacizumab OD 2×; OS 3×,	complete	Cort, MTX	no	Yes (stopped on his own)	0	no
2	74	m	positive	47	01/17	Occlusive OD + OS	0.3	0.6	0.4	0.4	yes OD	yes	OD: Triamcinolon parabulbar 2×; Ozurdex 3×,	Incomplete (persisting CME)	Cort, AZA, MTX	no	Yes (persisting macular edema)	2.5	ADA, MTX
3	49	m	positive	25	05/17	Non-occlusive OD + OS	1.4	0.3	2.3	0.4	no	no	no	complete	Cort, AZA	no	no	5	AZA
4	40	m	positive	19	01/12	Non-occlusive OD + OS	0	0	0	0	yes	no	no	complete	Cort	no	no	2	no
5	56	m	positive	35	03/17	Occlusive OD	0.4	0	0.5	0.1	no	no	no	complete	Cort	no	no	5	no
6	39	w	positive	33	03/16	Occlusive OS	0	0	0	0	no	no	no	complete	Cort, AZA	Pressure in the throat (once)	no	0	AZA
7	80	m	positive	37	03/17	Occlusive OD >> OS	0.1	1	0.1	0.4	yes OS	no	OS Ozurdex 2×, Illuvien 1×, (06/20)	complete	Cort, ADA, MTX	Herpetic infection	no	0	ADA, MTX
8	74	w	positive	32	11/18	Occlusive OS	0	0.3	1.4	0.1	yes OS	no	Ranibizumab 9×,	complete	Cort, AZA, MTX	Elevated liver encymes	Yes (adverse events)	0	MTX
9	37	w	positive	8	02/18	Occlusive OD + OS	0	0.2	0.1	0.1	yes OS	no	no	complete	Cort	no	no	5	no
10	58	m	positive	32	01/17	Occlusive OD + OS	0.2	0	0	0	no	no	no	complete	Cort, AZA, MTX	Hair loss	Yes (adverse events)	10	MTX
11	34	w	positive	31	12/17	Occlusive OD + OS	0.2	0.3	0	0	yes OS	no	no	complete	ADA	no	no	0	ADA
12	41	w	positive	55	12/15	Non-occlusive OD + OS	0	1	0	0.5	yes OD	no	no	complete	Cort, MTX, AZA	no	Yes (adverse events)	0	AZA
13	37	m	positive	49	11/16	Occlusive OD + OS	0.4	0	0.1	0	yes OD	Recurrence OD	OD Ozurdex, Bevacizumab 2×; OS Bevacizumab 4×,	Incomplete (persisting CME)	Cort, AZA, ADA, MTX	no	Yes (inefficacy)	5	ADA, MTX
no	36	w	positive	23	11/18	Occlusive OS	0	0	0	0	no	no	no	complete	Cort, ADA, MTX	no	no	0	ADA, MTX
15	69	m	positive	14	07/19	Occlusive OD + OS	0.7	1.3	0.2	0.2	no	no	no	complete	Cort, MTX	no	no	7.5	MTX
16	83	m	positive	6	07/20	Non-occlusive OD + OS	0.5	1.3	0.3	1.2	no	no	Bevacizumab OD/OS 2×,	Ongoing (less than 6 month therapy)	Cort, MTX	no	no	7.5	MTX
17	67	m	positive	11	05/19	Non-occlusive OD + OS	1.6	1.3	2	1.5	yes	no	Triamcinolon parabulbar	Complete (secondary glaucoma, corneal decompensation)	Cort CyA	no	no	0	no

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
