# Peer review of "Treatment in Latent Tuberculosis Uveitis—Is Immunosuppression Effective or Is Conventional 3- or 4-Drug Antituberculosis Therapy Mandatory?"

_jcm, 2022, doi:10.3390/jcm11092419_

Round 1
Reviewer 1 Report
The novelty and significance of manuscipt is good, but there are some quesetions:
1) Problem of patient criteria: excluding other infected uveitis should not be judged as latent tuberculosis uveitis by quantiferon elisa test results alone. And T-spot is relatively more accurate.
2) Uveitis treatment methods are randomly combined on a conventional basis. For uveitis without a clear cause, hormone therapy is the normalization, and the article is not different from the traditional treatment methods. When there are special cases, whether to use other treatment, the specific reasons or choice methods should be given. The treatment of isoniazid prevention is not specified in the relevant treatment guidelines
3) The observation objectives of the article do not reflect the theme of this study, uveitis without obvious infection itself does not require conventional antituberculosis drug treatment, and this article cannot provide effective evidence that this innovative treatment method can effectively improve the effectiveness of uveitis treatment and inhibit tuberculosis recurrence.
Author Response
Thank you very much for the comments. We revised parts of the manuscript (esp. in the chapters Introduction and Discussion) for clarifying this issue. Regarding Comment 1.: We revised the manuscript and updated the paragraph "methods". Regarding Comment 2.: We revised the paragraph Immunosuppression to clarify the treatment for each patient and also included a table with the therapy for each patient. Regarding Comment 3.: Our patients were diagnosed as having LTBI in case of a positive Quantiferon test (and a negative chest x-ray or ct) and after ruling out other non-infectious and infectious cases of uveitis. The actual literature recommend in uveitis patients having LTBI to apply conventional (3- or 4-drug) ATT as treatment strategy, sometimes in addition with systemic corticosteroids. We did not apply conventional ATT in our patient group. Due to the immunogenic nature of uveitis in LTBI our patients received different immunosuppressives - 11 patients with and 6 patients without INH-prophylaxis. After a median follow-up of 28months none of our patients had TB activation and only 29% of our patients had recurrence of uveitis. In the mentioned ATT-treatment studies recurrence rates were much higher - up to 75% of patients had uveitis relapses after ATT-treatment. In addition the side effect spectrum of ATT is worther than with immunosuppressive therapy.Reviewer 2 Report
TB-associated uveitis is a major cause for uveitis in tuberculosis endemic countries, and posterior uveitis is the most common manifestation. It’s disputable in terms of appropriate treatment for uveitis patients with LTBI, immunosuppression alone or combined with anti-tuberculosis therapy. In this study, the author performed a retrospective analysis of posterior patients with LTBI and concluded that additional ATT is not necessary for uveitis treatment in patients with latent tuberculosis. The medical history of patients is well-preserved and statistical method is reliable. Overall, this study is well-organized and instrumental for clinical practice.
Author Response
Thank you very much for the comments. We revised parts of the manuscript (esp. in the chapters Introduction and Discussion) to further improve the paper.Round 2
Reviewer 1 Report
The author has solved my problem and I have no problems.